# Dry Eye Following Femtosecond Laser-Assisted Cataract Surgery: A Meta-Analysis

**DOI:** 10.3390/jcm11216228

**Published:** 2022-10-22

**Authors:** Wei-Tsun Chen, Yu-Yen Chen, Man-Chen Hung

**Affiliations:** 1Department of Ophthalmology, Taichung Veterans General Hospital, Taichung 407, Taiwan; 2School of Medicine, National Yang Ming Chiao Tung University, Taipei 112, Taiwan; 3Wilmer Eye Institute, Johns Hopkins University School of Medicine, Baltimore, MD 21287, USA; 4Institute of Public Health and Community Medicine Research Center, National Yang Ming Chiao Tung University, Taipei 112, Taiwan; 5School of Medicine, Chung Shan Medical University, Taichung 402, Taiwan; 6Department of Post-Baccalaureate Medicine, College of Medicine, National Chung Hsing University, Taichung 402, Taiwan; 7Department of Medical Education, Taichung Veterans General Hospital, Taichung 407, Taiwan

**Keywords:** dry eye, femtosecond laser-assisted cataract surgery (FLACS), phacoemulsification, cornea

## Abstract

This study investigates the dry eye effect after femtosecond laser-assisted cataract surgery (FLACS) and also compares the risk of postoperative dry eye between FLACS and manual cataract surgery (MCS). We searched various databases between 1 January 2000 and 15 October 2022 and included peer-reviewed clinical studies in our review. Dry eye parameters were extracted at baseline and postoperative day one, week one, one month, and three months. Parameters included were the ocular surface discomfort index (OSDI), tear secretion (tear meniscus height, Schirmer’s test), microscopic ocular surface damage (fluorescein staining), and tear stability (first and average tear breakup time). Additionally, the differences of each parameter at each time point were compared between FLACS and MCS. In total, six studies of 611 eyes were included. On postoperative day one, increased, pooled standardised mean differences (SMDs) were noted in the OSDI, tear secretion, tear film instability, and microscopic damage. During postoperative week one, dry eye worsened. Fortunately, dry eye achieved resolution afterwards and nearly returned to the baseline level at postoperative three months. When the parameters were compared between FLACS and MCS, those of FLACS had higher severities, but most were not statistically significant. Dry eye impact was approximately the same in FLACS and MCS at postoperative three months.

## 1. Introduction

Dry eye is a common postoperative complaint from patients who underwent manual cataract surgery (MCS) with conventional phacoemulsification [1,2]. Symptoms include foreign body sensation, pain, blurred vision, ocular discomfort, burning, and dryness. These symptoms negatively affect patients’ satisfaction with surgery, quality of life, and burden public health [3]. After cataract surgery, signs of dry eye include a decreased tear breakup time, decreased corneal sensitivity, and increased ocular surface staining [2,4,5]. The pathogenic factors consist of inflammation, microscopic damage, neurosensory destruction on the ocular surface, tear film instability, and hyperosmolarity [6,7,8].

Since 2010, the femtosecond laser has been used in cataract surgery. Femtosecond laser-assisted cataract surgery (FLACS) provides precise anterior capsulotomy, safe lens fragmentation, and accurate corneal incision. Thus, it uses less ultrasound energy and phacoemulsification time [9], possibly leading to less postoperative inflammation and less dry eye. However, direct contact of the ocular surface with the vacuum and sustained pressure of the suction ring during FLACS may cause hyperaemia and microscopic damage to the ocular surface. In addition, laser procedures in FLACS may potentially affect the tear film [10]. All these reasons may result in dry eye.

Previous studies comparing FLACS and MCS were primarily concerned with the refractory outcome (e.g., visual acuity and spherical equivalent) and complication rate (e.g., anterior capsule tear or posterior capsule rupture) [9,11,12,13,14,15,16]. However, very few studies have investigated post-FLACS dry eye or compared the risk of dry eye between the two surgery groups. Therefore, we conducted this meta-analysis to investigate the impact of FLACS on dry eye and then compared postoperative dry eye after FLACS and MCS.

## 2. Materials and Methods

### 2.1. Search Strategy

This study was conducted according to the preferred reporting items for systematic reviews and meta-analyses (PRISMA) guidelines. We searched the PubMed, EMBASE, and Cochrane databases for studies published from 1 January 2000 to 15 October 2022, using the keywords ‘femtosecond laser-assisted cataract surgery’ and ‘dry eye’. Studies were screened first by examining the titles and abstracts and then scrutinising full texts. Bibliographies were also manually searched for the relevant literature.

### 2.2. Inclusion and Exclusion Criteria

Only peer-reviewed journal articles were included. They should be original, prospective, or randomised control clinical studies investigating dry eye presentation after FLACS. Reviews, meta-analyses, or conference abstracts were excluded because of repeated data. Two researchers (W.-C. Chen and Y.-Y. Chen) independently assessed the articles. A third researcher (M.-C. Hung) intervened if consensus was not reached.

Evaluation of the quality of included articles was performed independently by two researchers (W.-C. Chen and Y.-Y. Chen) using ROBINS-I risk of bias assessment tool. A third researcher (M.-C. Hung) reassessed and made the final decision if discrepancies occurred. ROBINS-I assesses the risk of bias in 7 domains, including confounding, selection of participants, classification of interventions, deviations from intended interventions, missing data, measurement of outcomes, and selection of the reported result. Each domain contains a set of questions (criteria). The risk of bias judgement of each domain was categorised into ‘Low risk’, ‘Moderate risk’, ‘Serious risk’, and ‘Critical risk’ of bias. Then, the overall risk of bias was judged according to the assessment of each domain.

### 2.3. Data Extraction

The following data were tracked from each included article: the first author, year of publication, and number/age/gender of participants. We also recorded the baseline (preoperative) and postoperative parameters regarding dry eye with: the ocular surface disease index (OSDI), tear meniscus height, Schirmer’s test, fluorescein staining, first tear breakup time, and average tear breakup time.

### 2.4. Definitions of Parameters

The OSDI was adopted to evaluate dry eye symptoms. The questionnaire included 12 questions about eye discomfort, visual function, and environmental triggers. A higher OSDI implies more severe dry eye [17]. Tear meniscus height was assessed via corneal topography in order to measure the height of the inferior tear meniscus [18]. A lower tear meniscus height implies a sign of dry eye. Schirmer’s test, also an index of tear secretion, was performed with sterile strips inserted at the lateral third of the lower eyelid margin [19]. The strips were removed five minutes later and the amount of wetting of the paper strips was measured. A lower Schirmer score suggests the diagnosis of dry eye. Fluorescein staining was applied to assess ocular surface damage [20]. Topical fluorescein readily enters and stains the corneal stroma where the epithelium is absent or when the epithelial cells have lost intercellular junctions. A higher score of fluorescein staining is a sign of dry eye. Tear film breakup time is a clinical evaluation of evaporative dry eye disease. Further, it is performed by instilling topical fluorescein into the eyes [21]. The number of seconds that elapsed between the last blink and the appearance of the first dry spot in the tear film was recorded as the first tear breakup time. Similarly, the average tear breakup time was recorded. A higher tear breakup time indicates tear film instability.

### 2.5. Statistical Analysis

Meta-analysis was performed using the Comprehensive Meta-Analysis software, version 3 (Biostat, Englewood, NJ, USA). First, we calculated the standardised mean differences (SMDs) of each index between the post-FLACS time points and baseline. The SMD from each study was computed by dividing the mean difference between each time point and baseline by the standard deviation in order to ensure that the difference was on the same scale. Then, the SMDs were pooled to derive the overall differences between post-FLACS and baseline according to each time point. Second, we compared the differences between the FLACS and MCS groups. The SMDs from each study were pooled to derive the overall values using a similar algorithm. Thus, we could then know which surgery was favoured. The heterogeneity among the studies was determined using the *I*^2^ statistic, and an *I*^2^ statistic of ≥50% would represent high heterogeneity. Funnel plots and Egger’s test were used to assess publication bias.

## 3. Results

### 3.1. Search Results

The PRISMA flow diagram is shown in Figure 1. A total of 67 studies were identified initially. After eliminating duplicated articles (*n* = 8), we removed non-relevant studies by screening titles and abstracts (*n* = 52). Then, a full-text review was performed. Conference abstracts were excluded (*n* = 1). Finally, six studies were enrolled in our meta-analysis [22,23,24,25,26,27].

### 3.2. Evaluation of the Quality of Included Studies

Risk of bias for each study assessed by the ROBINS-I tool is presented in Table 1. The overall results showed that one study (Schargus) had low risk of bias, four studies (Yu, Shao, Zhou, and Xu) had moderate risk of bias, and one study (Ju) had severe risk of bias. None of them had critical risk of bias.

### 3.3. Characteristics of Included Studies

The characteristics of the studies included in the meta-analysis are presented in Table 2. A total of 678 eyes from 611 patients were enrolled in six studies, with 359 eyes receiving FLACS and 319 eyes receiving MCS. Of the included studies, two were randomised controlled trials and four were prospective cohort studies. Five studies were conducted in China, whereas one study was performed in Germany. The mean age of the participants was 60 to 70 years in most studies.

### 3.4. Outcome Assessment of FLACS Group

Table 3 presents the three parameters (OSDI, tear meniscus height, and Schirmer’s test) at baseline and postoperative time points. Table 4 shows the values of the other three parameters (fluorescein staining, first tear breakup time, and average tear breakup time). The postoperative time points include day one, week one, one month, and three months.

The FLACS group pooled analyses comparing the postoperative and baseline values of the six parameters are presented in Figure 2 and Figure 3. The overall SMDs showed increased values at postoperative day one in four of the six parameters (OSDI, tear meniscus height, Schirmer’s test, and fluorescein staining). The increase was statistically significant in tear meniscus height (SMD: 0.456, 95% confidence interval (CI): 0.257 to 0.655), Schirmer’s test (SMD: 0.132, 95% CI: 0.037 to 0.226) and fluorescein staining (SMD: 3.550, 95% CI: 0.354 to 6.747), but was not statistically significant in OSDI (SMD: 5.610, 95% CI: −2.191 to 13.411). Subsequently, tear meniscus height and Schirmer’s test scores decreased to a level lower than baseline, while OSDI and fluorescein staining scores remained higher than baseline. The SMDs of each parameter had a tendency toward zero over time.

Regarding the first and average tear breakup times, both had lower values than baseline from postoperative day one to the first month. The decreased values were only significant in the first tear breakup time at postoperative week one and the first month. The SMDs of the first and average tear breakup times trended towards zero with time. Finally, at postoperative three months, the six parameters were nearly similar to their baseline values except for tear meniscus height, which was significantly lower than at baseline (SMD: −0.172, 95% CI: −0.328 to −0.015).

### 3.5. Outcome Assessment Comparing FLACS and MCS Group

Figure 4 and Figure 5 compare the postoperative change in six parameters between FLACS and MCS at various postoperative time points. The FLACS group had a higher reduction in tear meniscus height, Schirmer’s test, fBUT, and avBUT. In addition, it had a higher increase in OSDI and fluorescent staining than the MCS group at every postoperative time point. In addition, the FLACS group showed less tear secretion postoperatively. However, most differences between FLACS and MCS were becoming less from postoperative day one to three months. Further, the differences were only significant at the following three time points: Schirmer’s test at postoperative day one (SMD: −0.208, 95% CI: −0.397 to −0.020), one month (SMD: −0.309, 95% CI: −0.534 to −0.085), and first tear breakup time at postoperative week one (SMD: −0.685, 95% CI: −1.058 to −0.311).

### 3.6. Heterogeneity and Publication Bias

Most analyses showed high between-study heterogeneity when evaluating the SMDs of six parameters (*I*^2^ > 75%). Concerning publication bias, Figure 6 demonstrates the funnel plots of studies regarding the post-FLACS effects. Regarding OSDI, tear meniscus height and Schirmer’s test, the *p*-values of the Egger’s test were 0.31, 0.94, and 0.65, respectively—revealing no significant publication biases. Significant publication biases were noted regarding post-FLACS effects corresponding to fluorescent staining, first tear breakup time, and average breakup time (all Egger’s tests *p* < 0.01).

Funnel plots of the studies comparing postoperative effects between FLACS and MCS are presented in Figure 7. They exhibited no significant publication biases in all six parameters of dry eye symptoms/signs (all Egger’s tests *p* > 0.1).

Since the publication bias is statistically significant regarding post-FLACS impacts on fluorescein staining, fBUT, and avBUT, we applied the trim-and-fill method to deal with the publication biases. After trimming the studies that caused a funnel plot’s asymmetry and filling imputed missing studies in the funnel plot based on the bias-corrected overall estimate, the funnel plots were adjusted and are presented in Figure 8. The direction and significance of SMD did not change after adjusting the publication biases. Therefore, our previous statistical analyses regarding SMD were convincible.

## 4. Discussion

This meta-analysis included six studies focusing on dry eye after FLACS. Six parameters (OSDI, tear meniscus height, Schirmer’s test, fluorescein staining, first breakup time, and average breakup time) were used to evaluate dry eye symptoms/signs, which were also compared between FLACS and MCS groups. On postoperative day one, eyes receiving FLACS had transiently increased dry eye symptoms (OSDI) and tear secretion (tear meniscus height and Schirmer’s test) but then decreased. Microscopic ocular surface damage (fluorescein staining) was significantly increased on postoperative day one and week one but improved after one month. Tear film instability (first breakup time and average breakup time) lasted for one month after surgery and then returned to the baseline level. Three months after surgery, only tear meniscus height was significantly decreased, while all the other parameters were similar to baseline. Compared with MCS, FLACS had a greater tendency towards dry eye in the early postoperative stage. However, the dry eye symptoms/signs between FLACS and MCS showed no significant differences three months after surgery.

This study is the first meta-analysis to compare the impact on postoperative dry eye between FLACS and MCS, to the best of our knowledge. In our study, a transient increase in tear secretion on postoperative day one may be related to surgical-induced pain. One possible explanation for the tear film instability presenting itself immediately after surgery is inflammation. Wound epithelial cells secrete inflammatory factors that accumulate in tears. The bandage of the eye decreases the tear removal rate and aggravates the inflammatory reaction, hyperosmolarity in tears, and subjective discomfort.

Regarding microscopic ocular surface damage, multiple reasons are responsible, including preoperative instillation of povidone-iodine and local anaesthesia [28,29], intraoperative irrigation, and light exposure [30]. Dry eye symptoms improved, but signs were worse at postoperative week one, implying more cytokines were released from the wound in order to induce inflammation. In addition, our study found that FLACS had a more severe effect on dry eye than MCS. This effect may be due to the suction ring in FLACS, injuring the limbal stem cells, conjunctival epithelium, and goblet cells. It is similar to the dry eye mechanism after laser-assisted in situ keratomileusis [31]. In addition, the extra laser procedure in FLACS leads to prolonged light exposure, thereby deteriorating tear film stability.

Fortunately, in our study, the symptoms/signs of dry eye immediately following FLACS almost returned to baseline within three months postoperatively. This result might be explained by the anti-inflammatory effects of postoperative eye drops. Previous studies have revealed that neuroregeneration occurs 25 days postoperatively [32], supporting our finding that postoperative dry eye tends to improve. Furthermore, the differences in dry eye parameters between FLACS and MCS mainly have no significant difference and have a decreasing trend. However, Yu et al. have found that FLACS causes more ocular surface damage than MCS in patients with pre-existing dry eye [22]. Therefore, preoperative screening and postoperative treatment for dry eye should be performed meticulously for those receiving FLACS with a pre-existing unhealthy ocular surface.

The main limitation of our meta-analysis is the heterogeneity among the included studies. The between-study variations may arise from differences in surgical machines, study protocols, inclusion criteria, and perioperative use of topical medication. Five of the six enrolled studies used the LenSx femtosecond laser system (Alcon Laboratories, Fort Worth, TX, USA). Only Schargus et al. used the CATALYS laser system (Johnson and Johnson, New Brunswick, NJ, USA). Different docking devices used in the laser platforms may cause different effects on the ocular surface [24,33]. Another limitation is that most included studies have a non-randomised design, increasing bias.

Moreover, the parameters used in our meta-analysis (OSDI, tear meniscus height, Schirmer’s test, fluorescein staining, and tear breakup time) are not objective enough and are prone to observers’ errors. Previous studies have suggested that tear film osmolarity and matrix metalloproteinase levels are more reliable dry eye tests and correlate well with dry eye severity [20,34,35]. In addition, meibomian gland dysfunction, lipid layer thickness, inflammatory levels, and goblet cell densities also play an important role in dry eye [36,37]. These parameters should be assessed in further studies. Still another limitation is that we cannot perform subgroup analyses according to cataract grading or phacoemulsification time, which are relevant with post-operative dry eye. We have extracted data of cataract grading from three studies (Yu, Zhou, and Xu) and phacoemulsification time from two studies (Yu and Xu). However, the information was presented as overall proportion or mean, without mentioning the individual dry eye symptoms/signs corresponding to each category of cataract grading or phacoemulsification time. The lack of details and the too few study numbers makes subgroup analyses infeasible.

The strength of our study is that our results provide an evaluation of dry eye symptoms/signs following FLACS and include comparisons with those following MCS. Therefore, we could have a better understanding of postoperative dry eye risk. More comprehensive studies will need to be conducted, thereby supplying evidence for further meta-analyses.

## 5. Conclusions

In conclusion, both FLACS and MCS can induce dry eye. The adverse effects of FLACS on the ocular surface are more severe in FLACS than in MCS. Fortunately, these effects are transient and are resolved within three months after surgery. Cataract surgeons should select FLACS candidates carefully and adopt preoperative evaluation and postoperative therapy for dry eye. Further studies are warranted to verify and understand the post-FLACS dry eye mechanism.

## Figures and Tables

**Figure 1 jcm-11-06228-f001:**
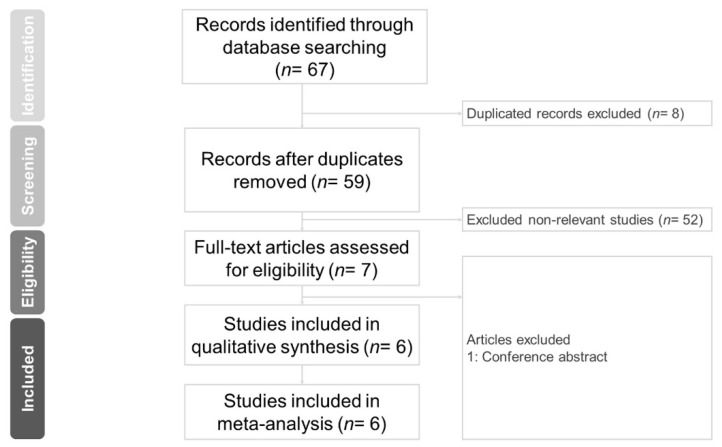
Preferred reporting items for systemic reviews and meta-analyses (PRISM) flow diagram for searching and identifying included studies.

**Figure 2 jcm-11-06228-f002:**
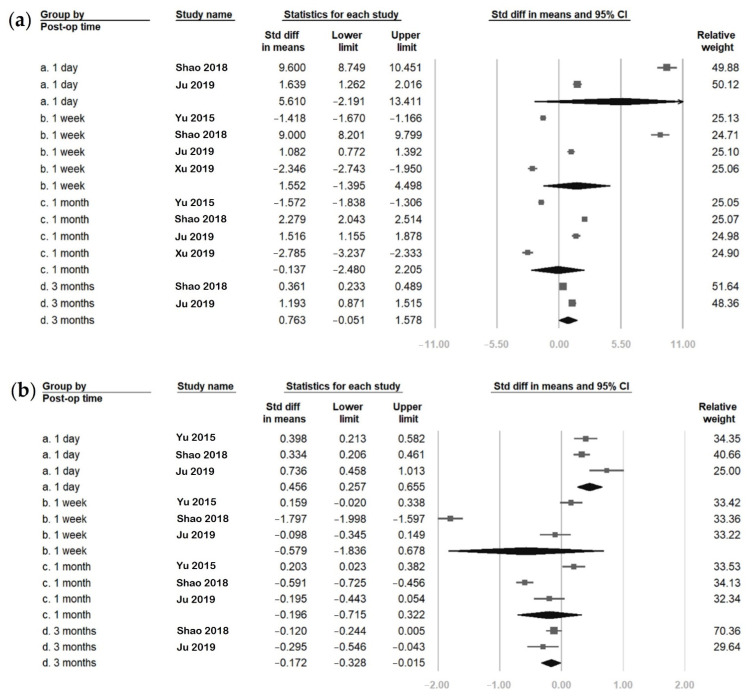
Overall effect of femtosecond laser-assisted cataract surgery (FLACS) on (**a**) ocular surface disease index (OSDI), (**b**) tear meniscus height, and (**c**) Schirmer’s test. The square represents the standardised mean difference of each study. The size of square stands for the relative weight of each study. The lozenge represents the overall standardised mean difference.

**Figure 3 jcm-11-06228-f003:**
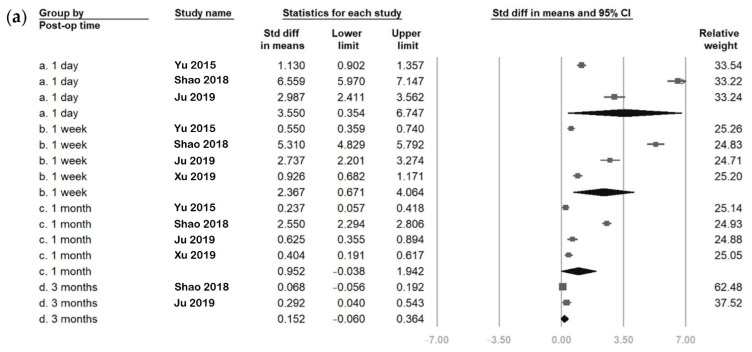
Overall effect of femtosecond laser-assisted cataract surgery (FLACS) on (**a**) fluorescein staining, (**b**) first tear breakup time (fBUT), and (**c**) average tear breakup time (avBUT). The square represents the standardised mean difference of each study. The size of square stands for the relative weight of each study. The lozenge represents the overall standardised mean difference.

**Figure 4 jcm-11-06228-f004:**
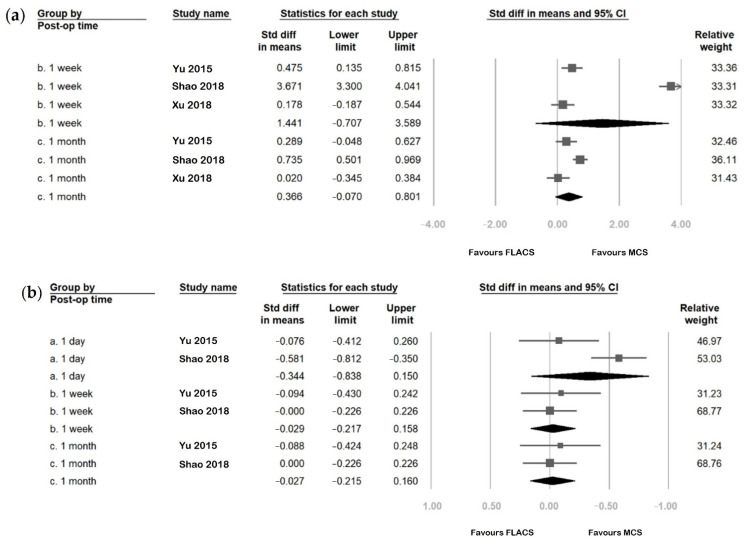
Comparison of (**a**) ocular surface disease index (OSDI), (**b**) tear meniscus height, and (**c**) Schirmer’s test between the femtosecond laser-assisted cataract surgery (FLACS) group and manual cataract surgery (MCS) group. *I*^2^ represents heterogeneity. The square represents the standardised mean difference of each study. The size of square stands for the relative weight of each study. The lozenge represents the overall standardised mean difference.

**Figure 5 jcm-11-06228-f005:**
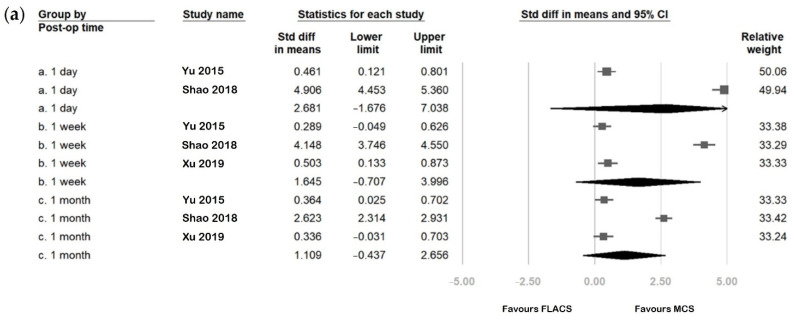
Comparison of (**a**) fluorescein staining, (**b**) first tear breakup time (fBUT), and (**c**) average tear breakup time (avBUT) between the femtosecond laser-assisted cataract surgery (FLACS) group and manual cataract surgery (MCS) group. *I*^2^ represents heterogeneity. The square represents the standardised mean difference of each study. The size of square stands for the relative weight of each study. The lozenge represents the overall standardised mean difference.

**Figure 6 jcm-11-06228-f006:**
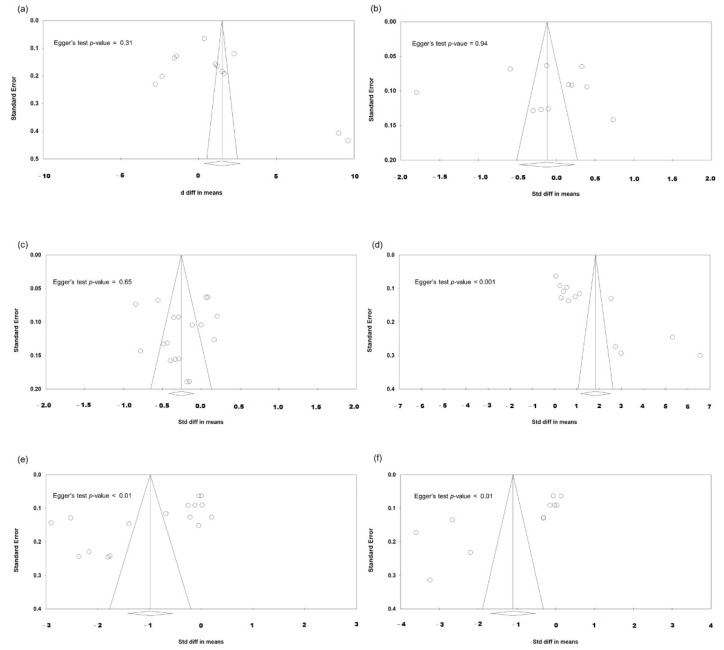
Funnel plots evaluating the publication biases regarding post-FLACS impacts on the six dry eye parameters (**a**) OSDI, (**b**) tear meniscus height, (**c**) Schirmer’s test, (**d**) fluorescein staining, (**e**) fBUT, and (**f**) avBUT. The lozenge stands for overall standardised mean difference.

**Figure 7 jcm-11-06228-f007:**
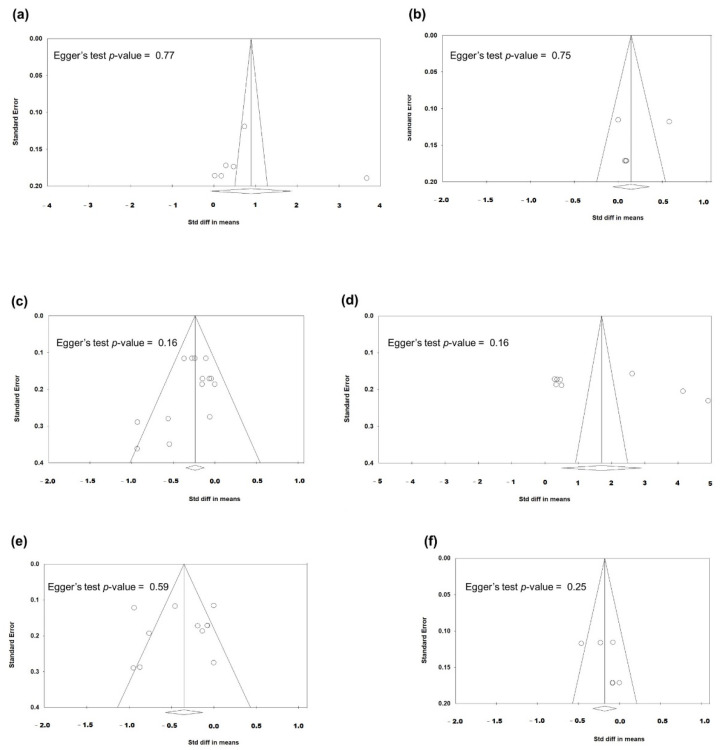
Funnel plots evaluating the publication biases regarding the comparison between FLACS and MCS on the six dry eye parameters (**a**) OSDI, (**b**) tear meniscus height, (**c**) Schirmer’s test, (**d**) fluorescein staining, (**e**) fBUT, and (**f**) avBUT. The lozenge stands for overall standardised mean difference.

**Figure 8 jcm-11-06228-f008:**
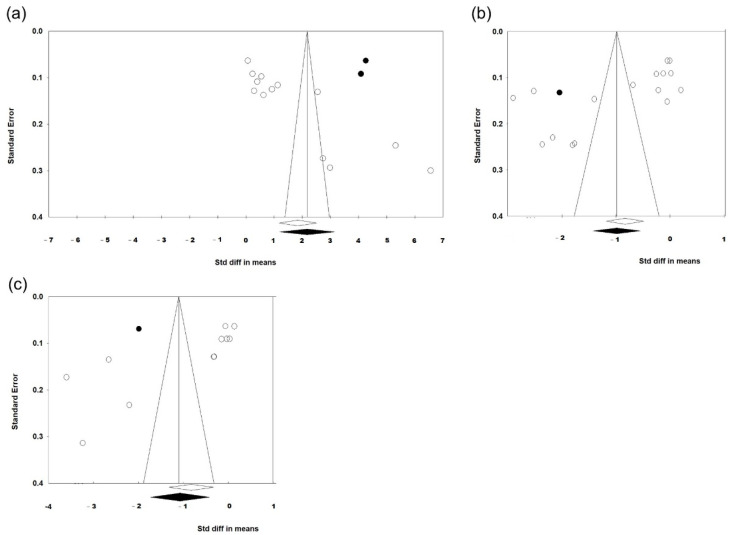
Funnel plots after using trim-and-fill method to adjust the publication biases regarding post-FLACS impacts on the dry eye parameters (**a**) fluorescein staining, (**b**) fBUT, and (**c**) avBUT. The lozenge stands for overall standardised mean difference. The data points for imputed studies are highlighted in black.

**Table 1 jcm-11-06228-t001:** Risk of bias assessment for the individual studies included in the meta-analysis.

	D1	D2	D3	D4	D5	D6	D7	Overall
Yu [22]	Moderate	Low	Low	Low	Low	Moderate	Low	Moderate
Shao [23]	Low	Low	Low	Low	Low	Moderate	Moderate	Moderate
Schargus [24]	Low	Low	Low	Low	Low	Low	Low	Low
Ju [25]	Moderate	Low	Low	Low	Severe	Moderate	Moderate	Severe
Zhou [26]	Moderate	Low	Low	Low	Low	Moderate	Moderate	Moderate
Xu [27]	Moderate	Low	Low	Low	Low	Moderate	Moderate	Moderate

D1 = Bias due to confounding; D2 = bias in selection of participants into the study; D3 = bias in classification of interventions; D4 = bias due to deviations from intended interventions; D5 = bias due to missing data; D6 = bias in measurement of outcomes; D7 = bias in selection of the reported result; Low = low risk of bias; Moderate = moderate risk of bias; and Severe = severe risk of bias.

**Table 2 jcm-11-06228-t002:** Characteristics of the studies included in meta-analysis.

Author	Year	Type	Country	Study Population	Num of Patients	Num of Eyes	Age, Year (Mean ± SD)	Male (*n*, %)	Cataract Grading	Phaco Time (s)
Yu [22]	2015	PCS	China	FLACS	73	73	69.0 ± 10.6	34 (46.6)	NS 1+ (24.7%), NS 2+ (53.4%), NS 3+ (17.8%), NS 4+ (4.1%)	35.5 ± 18.4
				MCS	64	64	71.8 ± 10.1	27 (42.2)	NS 1+ (23.4%), NS 2+ (54.7%), NS 3+ (18.8%), NS 4+ (3.1%)	46.7 ± 26.7
Shao [23]	2018	RCT	China	FLACS	123	150	65.7 ± 11.8	67 (44.7)	NR	NR
				MCS	110	150	69.1 ± 12.6	62 (41.3)	NR	NR
Schargus [24]	2020	RCT	Germany	FLACS	17	17	67.4 ± 9.7	7 (41.2)	NR	NR
				MCS	17	17	66.0 ± 7.5	9 (52.9)	NR	NR
Ju [25]	2019	PCS	China	FLACS	38	38	72.6 ± 8.7	16 (42.1)	NR	NR
Zhou [26]	2018	PCS	China	FLACS	26	26	63.2 ± 8.6	11 (42.3)	NS 1+ (0%), NS 2+ (38.5%), NS 3+ (61.5%), NS 4+ (0%)	NR
				MCS	27	27	60.6 ± 6.4	10 (37.0)	NS 1+ (0%), NS 2+ (40.7%), NS 3+ (59.3%), NS 4+ (0%)	NR
Xu [27]	2019	PCS	China	FLACS	55	55	64.5 ± 7.6	25 (45.5)	NS 1+ (20.0%), NS 2+ (36.4%), NS 3+ (30.9%), NS 4+ (12.7%)	37.7 ± 10.5
				MCS	61	61	63.2 ± 8.6	27 (44.3)	NS 1+ (23.0%), NS 2+ (34.4%), NS 3+ (31.1%), NS 4+ (11.5%)	48.0 ± 13.6

Num= number; PCS= prospective cohort study; RCT= randomised controlled trial randomised control trial; FLACS= femto-second laser cataract surgery; MCS = manual cataract surgery; NS = nuclear sclerotic cataract; NR = not reported; and Phaco = phacoemulsification.

**Table 3 jcm-11-06228-t003:** Post-operative changes in OSDI, tear meniscus height, and Schirmer’s test.

Study	Group	Num of Eyes	OSDI					Tear Meniscus Height					Schirmer’s Test				
			Baseline	1 Day	1 Week	1 Month	3 Months	Baseline	1 Day	1 Week	1 Month	3 Month	Baseline	1 Day	1 Week	1 Month	3 Months
Yu [22]	FLACS	73	22.9 ± 4.2	NR	11.0 ± 5.5	9.1 ± 6.0	NR	0.25 ± 0.12	0.32 ± 0.19	0.27 ± 0.13	0.28 ± 0.16	NR	9.2 ± 7.0	10.3 ± 8.5	7.2 ± 6.4	7.6 ± 7.2	NR
	MCS	64	23.7 ± 5.8	NR	8.8 ± 4.9	8.0 ± 4.9	NR	0.24 ± 0.15	0.30 ± 0.17	0.25 ± 0.15	0.26 ± 0.14	NR	9.4 ± 7.4	11.0 ± 8.6	7.3 ± 6.3	8.6 ± 6.9	NR
Shao [23]	FLACS	150	0.5 ± 0.2	5.3 ± 0.5	5.0 ± 0.5	2.2 ± 0.7	0.6 ± 0.3	0.37 ± 0.09	0.41 ± 0.13	0.22 ± 0.07	0.32 ± 0.05	0.36 ± 0.07	10.9 ± 4.1	11.3 ± 4.9	7.6 ± 3.7	8.8 ± 2.6	11.2 ± 5.0
	MCS	150	0.5 ± 0.4	4.0 ± 0.3	3.5 ± 0.6	1.8 ± 0.7	0.5 ± 0.4	0.35 ± 0.08	0.44 ± 0.11	0.20 ± 0.06	0.30 ± 0.06	0.37 ± 0.06	9.4 ± 4.0	10.7 ± 3.7	7.2 ± 3.3	8.0 ± 2.7	10.1 ± 5.4
Schargus [24]	FLACS	17	NR	NR	NR	NR	NR	NR	NR	NR	NR	NR	13.5 ± 7.9	NR	NR	12.3 ± 7.9	12.0 ± 8.3
	MCS	17	NR	NR	NR	NR	NR	NR	NR	NR	NR	NR	12.7 ± 8.2	NR	NR	14.9 ± 8.2	17.2 ± 8.7
Ju [25]	FLACS	38	8.4 ± 2.1	17.5 ± 5.5	16.0 ± 6.7	13.5 ± 3.6	11.7 ± 3.0	0.32 ± 0.11	0.41 ± 0.13	0.31 ± 0.07	0.30 ± 0.09	0.29 ± 0.07	12.9 ± 3.2	13.4 ± 2.6	10.6 ±2.3	11.4 ± 3.0	11.6 ± 2.6
Zhou [26]	FLACS	26	NR	NR	NR	NR	NR	NR	NR	NR	NR	NR	12.8 ± 1.9	NR	12.1 ±1.5	12.2 ± 2.2	12.2 ± 1.7
	MCS	27	NR	NR	NR	NR	NR	NR	NR	NR	NR	NR	13.5 ± 2.5	NR	11.9 ±1.5	11.3 ± 1.4	13.0 ± 2.1
Xu [27]	FLACS	55	24.5 ± 6.5	NR	10.4 ± 4.2	7.8 ± 4.4	NR	NR	NR	NR	NR	NR	9.4 ± 4.8	NR	9.4 ± 4.0	8.9 ± 3.7	NR
	MCS	61	24.8 ± 7.5	NR	11.6 ± 5.6	8.2 ± 4.9	NR	NR	NR	NR	NR	NR	8.7 ± 4.4	NR	8.7 ± 3.5	8.7 ± 3.3	NR

All data are displayed as mean ± SD. Num = number; OSDI = ocular surface disease index; FLACS = femtosecond laser-assisted cataract surgery; MCS = manual cataract surgery; and NR = not reported.

**Table 4 jcm-11-06228-t004:** Post-operative changes in fluorescein staining, and tear breakup time.

Study	Group	Num of Eyes	Fluorescein Staining					First Tear Breakup Time					Average Tear Breakup Time				
			Baseline	1 Day	1 Week	1 Month	3 Months	Baseline	1 Day	1 Week	1 Month	3 Month	Baseline	1 Day	1 Week	1 Month	3 Months
Yu [22]	FLACS	73	0.40 ± 0.52	1.46 ± 0.73	0.84 ± 0.53	0.59 ± 0.55	NR	5.5 ± 3.5	4.9 ± 3.4	4.4 ± 2.8	5.6 ± 3.9	NR	7.4 ± 4.3	7.2 ± 4.2	6.5 ± 3.3	7.7 ± 4.5	NR
	MCS	64	0.36 ± 0.49	1.13 ± 0.70	0.67 ± 0.65	0.39 ± 0.55	NR	5.0 ± 2.8	4.7 ± 3.5	4.6 ± 4.0	4.8 ± 3.4	NR	6.8 ± 4.3	7.1 ± 4.2	6.3 ± 4.6	7.1 ± 4.6	NR
Shao [23]	FLACS	150	0.46 ± 0.20	2.34 ± 0.31	1.88 ± 0.29	0.97 ± 0.20	0.51 ± 0.69	11.8 ± 0.8	8.5 ± 1.4	8.0 ± 1.4	11.7 ± 2.1	11.8 ± 2.8	12.7 ± 1.1	10.0 ± 0.8	9.0 ± 0.9	12.6 ± 1.7	12.9 ± 1.6
	MCS	150	0.38 ± 0.22	1.22 ± 0.28	1.02 ± 0.21	0.48 ± 0.14	0.46 ± 0.35	11.0 ± 1.2	8.2 ± 0.0	8.1 ± 1.1	10.9 ± 1.6	11.0 ± 2.1	13.2 ± 1.3	10.1 ± 0.8	9.3 ± 0.9	13.2 ± 1.8	13.4 ± 1.4
Schargus [24]	FLACS	17	5.14 ± 0.39	NR	NR	NR	NR	NR	NR	NR	NR	NR	NR	NR	NR	NR	NR
	MCS	17	5.57 ± 0.17	NR	NR	NR	NR	NR	NR	NR	NR	NR	NR	NR	NR	NR	NR
Ju [25]	FLACS	38	0.89 ± 0.73	4.13 ± 1.17	3.21 ± 0.91	1.34 ± 0.71	1.10 ± 0.77	10.7 ± 1.2	8.1 ± 1.2	7.0 ± 1.7	10.4 ± 1.5	11.1 ± 2.1	11.6 ± 1.0	9.4 ± 1.0	8.5 ± 0.9	11.3 ± 0.8	11.3 ± 0.9
Zhou [26]	FLACS	26	NR	NR	NR	NR	NR	14.3 ± 2.0	NR	10.2 ± 2.5	10.7 ± 2.0	14.2 ± 1.9	NR	NR	NR	NR	NR
	MCS	27	NR	NR	NR	NR	NR	14.4 ± 2.2	NR	8.8 ± 2.0	9.3 ± 1.9	14.3 ± 1.5	NR	NR	NR	NR	NR
Xu [27]	FLACS	55	0.55 ± 0.72	NR	1.38 ± 0.97	0.93 ± 1.02	NR	6.2 ± 2.0	NR	3.6 ± 1.6	4.8 ± 2.1	NR	NR	NR	NR	NR	NR
	MCS	61	0.51 ± 0.52	NR	1.01 ± 0.86	0.66 ± 0.89	NR	6.0 ± 1.6	NR	4.5 ± 2.0	4.8 ± 1.9	NR	NR	NR	NR	NR	NR

All data are displayed as mean ± SD. Num = number; FLACS = femtosecond laser-assisted cataract surgery; MCS = manual cataract surgery; and NR = not reported.

## Data Availability

The data from this study are available on request from the corresponding author.

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
