# Peer review of "Dry Eye Following Femtosecond Laser-Assisted Cataract Surgery: A Meta-Analysis"

_jcm, 2022, doi:10.3390/jcm11216228_

Round 1
Reviewer 1 Report
assisted cataract surgery: A meta-analysis. I found it an interesting and well written study.
In my opinion this article deserves to be published. Authors did a great job analysing an important aspect in the most frequent surgery in our job, the cataract surgery.
Although FLACS have demonstrated to be a well option, safe and effective procedure. We still need to perform several research to demonstrate that this technology is better than normal phacoemulsification technique. In this paper, the authors showed us that regarding dry eye, FLACS could have a no significant more impact in our patients, especially in the early postoperative stage.
Moreover, author recognize the limitations of their study as the heterogeneity and few studies included.
Minor changes:
I would like to recommend not to use ‘’PHACO’’ and use manual cataract surgery (MCS)
Author Response
Reviewer 1
Minor changes:
I would like to recommend not to use ‘’PHACO’’ and use manual cataract surgery (MCS)
Answer: Thank you for your comment. According to your comment, I have used ‘manual cataract surgery (MCS)’ instead of ‘PHACO’.
Reviewer 2 Report
In this manuscript, Chen et al. conducted a meta-analysis on dry eye after FLACS by synthesizing six objective parameters. The study also compared the dry eye signs between FLACS and Phaco, which suggested that FLACS had a greater tendency towards dry eye in the early postoperative stage. Although the I find this topic relevant and the manuscript well-written, there are several issues that need to be addressed, please find my comments below.
General comments
1. One of the major concerns is that the literature search is until 28 Feb 2022 and an updated search would be preferable to implement the new data.
2. Please perform risk of bias assessment for the included studies according to the criteria of meta-analysis.
Specific comments
1. Since nuclear sclerosis and surgery time are relevant with the dry eye symptoms, I suggest include a column describing the cataract grading or surgery parameters. It may provide more robust evidence to conduct subgroup analyses according to the nuclear hardness.
2. Line 55: “compare” should be “compared”.
3. Line 143-144: “OSDI (SMD: 5.610, 95% confidence interval (CI): −2.191 to 13. 411)” was not statistically significant. Please rephrase this sentence.
4. Figure 4 & Figure 5: I suggest to arrange “Favors FLACS” and “Favors Phaco” in the same order in each figure.
5. Line 176: Heterogeneity and publication bias. The heterogeneity of each analysis was not shown in the figures or in the manuscript. Please provide the detailed information.
6. Publication bias: Since significant publication biases were noted, I suggest to perform trim and fill analysis to recalculate the results.
